# Tell What You Hear From What You See - Video to Audio Generation Through Text

Xiulong Liu [*]               Kun Su [*]               Eli Shlizerman [†][*][‡]

## Abstract

The content of visual and audio scenes is multi-faceted such that a video stream can be paired with various audio streams and vice-versa. Thereby, in video-to-audio generation task, it is imperative to introduce steering approaches for controlling the generated audio. While Video-to-Audio generation is a well-established generative task, existing methods lack such controllability. In this work, we propose *VATT*, a multi-modal generative framework that takes a video and an optional text prompt as input, and generates audio and optional textual description (caption) of the audio. Such a framework has two unique advantages: i) Video-to-Audio generation process can be refined and controlled via text which complements the context of the visual information, and ii) The model can suggest what audio to generate for the video by generating audio captions. VATT consists of two key modules: *VATT Converter*, which is an LLM that has been fine-tuned for instructions and includes a projection layer that maps video features to the LLM vector space, and *VATT Audio*, a bi-directional transformer that generates audio tokens from visual frames and from optional text prompt using iterative parallel decoding. The audio tokens and the text prompt are used by a pretrained neural codec to convert them into a waveform. Our experiments show that when VATT is compared to existing video-to-audio generation methods in objective metrics, such as VGGSound audio-visual dataset, it achieves competitive performance when the audio caption is not provided. When the audio caption is provided as a prompt, VATT achieves even more refined performance (with lowest KLD score of 1.41). Furthermore, subjective studies asking participants to choose the most compatible generated audio for a given silent video, show that VATT Audio has been chosen on average as a preferred generated audio than the audio generated by existing methods. VATT enables controllable video-to-audio generation through text as well as suggesting text prompts for videos through audio captions, unlocking novel applications such as text-guided video-to-audio generation and video-to-audio captioning.

## 1   Introduction

The combination of human perception and cognition represents a "multi-modal" way of processing and interpreting scenes. For example, when we are presented with a silent video of a fountain show attended by a crowd of people gathered around the spectacle our interpretation might translate the visual scene into an auditory experience, where the visuals are semantically processed and transformed into a corresponding sound narrative in our mind. Thus, we may associate audio that mixes sounds of splashing water accompanied by people talking and laughing with possibly background music in sync with the fountain.

[*]Department of Electrical & Computer Engineering, University of Washington, Seattle, USA.
[†]Department of Applied Mathematics, University of Washington, Seattle, USA
[‡]Corresponding author: shlizee@uw.edu

38th Conference on Neural Information Processing Systems (NeurIPS 2024).

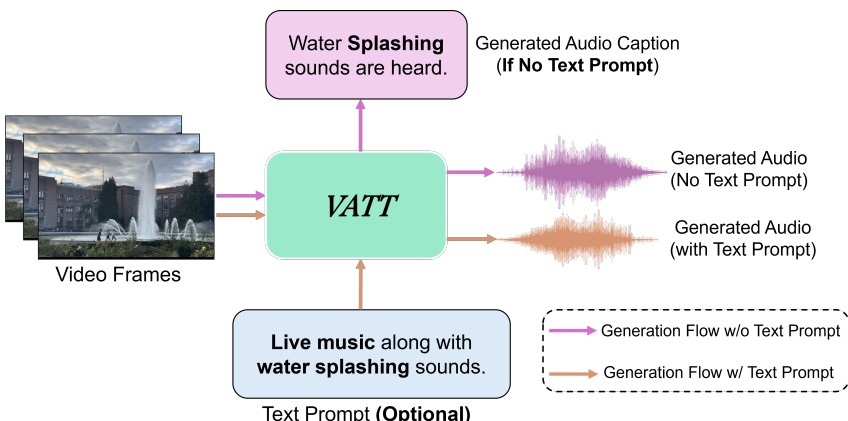

Figure 1: VATT is a flexible audio generative model capable of generating audio in two modes: i) When a silent video is the sole input, the model generates the audio along with a caption describing the possible audio that could match the video. ii) When in addition to the video, a text prompt is provided, the model generates audio aligned with both the video and the given text prompt.

As generative AI continues to progress, the incorporation of the aforementioned aspects into generative platforms presents itself as the future desirable capability. In particular, the goal of an ideal video-to-audio generative model would be to generate sounds that seamlessly match the video temporally and fully capture the semantics. Moreover, it is desirable to control such a generation process towards the themes and sounds that match user preference. Recent state-of-the-art approaches have adopted two types of generative modeling techniques: auto-regressive token-based modeling and diffusion-based modeling. These methods enable end-to-end video-to-audio generation and are applicable across a wide variety of video and audio categories. However, while these methods are capable of capturing the general semantics of sound sources in videos, they often overlook the subtleties of the context. For example, in a video depicting two cats in a territorial dispute, the model might produce a calm, amiable meowing sound, which contradicts the contentious nature of the scene. This discrepancy mainly stems from the limitations of the vision encoder, which struggles to distinguish between varying sound properties emitted by identical sound sources across different contexts, due to an incomplete understanding of the entire scene. Second, these methods lack controllability since the generation is conditioned only on visual frames, without taking into account the context and interpretation of the sounds. While text-to-audio models could explicitly control the context of the sounds, such models are based on text only without incorporating the rich and dynamic context of visuals, which could significantly inform video and audio alignment. Indeed, text only generative outcomes often result in unmatched audio with the visual (e.g., temporal misalignment or semantic loss).

To solve the above challenges, we propose a novel framework, Video-to-Audio Through Text (VATT), that is able to generate audio from both video frames and an optional text prompt describing the expected sounds. VATT consists of two modeling stages: i) Video-to-Caption stage, which converts video features into an audio caption through a pretrained large language model (LLM) with a learnable projection layer. Through this cross-modal conversion, visual features that are relevant to audio concepts are extracted. These features are closely connected to audio-related tasks such as audio captioning and audio generation. ii) Video + Text to Audio stage, that generates audio conditioned on the hidden states extracted from the LLM in the prior modeling stage. At its core, the proposed model in this stage is a bi-directional transformer decoder that generates audio using a token-based representation similar to [1, 2]. To obtain the conditioning on the hidden states of the preceding component, the projected video features along with the optional text prompts are concatenated together and fed into the LLM in stage i), with the hidden states from the last layer extracted and attached to the audio tokens for the decoder. The decoder is trained using masked token modeling, where the objective is to predict masked audio tokens from unmasked ones at varying masking ratios. During inference, starting from all tokens being masked, an efficient parallel decoding algorithm is implemented which gradually unmasks multiple tokens in parallel based on video and text inputs until a stop condition is met. Finally, the generated tokens are converted into audio waveforms through a neural audio codec decoder.

We perform experiments with the proposed framework on existing large-scale audio-visual datasets such as VGGSound [3] and Audioset-2M [4]. To facilitate training and evaluation with text, we created "V2A Instruction", a large-scale synthetic audio captions corpus, by prompting LTU-13B, an existing Audio LLM [5], to generate audio descriptions for both datasets. Our experiments demonstrate that the proposed model and its training method achieve competitive performance in comparison to previous video-to-audio methods on both objective and subjective metrics. Furthermore, it is designed to enable a controllable generation that adheres to both the video inputs and the text prompts. Indeed when a text prompt is provided, our experiments show significant improvement in audio metrics that measure the match of the generated sounds to the video. In addition, when the text prompt is not provided, our method can generate reasonable audio captions, which can be utilized for a potential description of the video or classification of sounds for a given video. These capabilities hence make VATT a multifaceted single model able to perform both text-guided video-to-audio generation and video-to-audio captioning. To summarize our contributions:

- To our best knowledge, we propose a first-of-its-kind framework that enables both text-guided video-to-audio generation and video-to-audio captioning through the integration of LLM.
- We create a large-scale synthetic audio captions dataset that facilitates text-conditional training and generation.
- Our method achieves state-of-the-art video-to-audio generation performance when compared with existing methods and enables text-controllable generation. In particular, our text-guided model surpasses existing SOTA in terms of KLD score (with lowest KLD score of 1.41) by a significant margin.
- VATT generates audio in an efficient way - an order of magnitude faster than existing methods.

## 2 Related Works

### 2.1 Visual-to-Audio Generation

Visual-to-Audio Generation task has drawn significant attention since generative frameworks such as diffusion and transformer-based architectures have been developed. Existing Visual-to-Audio generation approaches can be divided into two branches of studies based on audio categories: *visual-to-music* generation and *visual-to-natural* sound generation. In visual-to-music generation domain, earlier studies explored Midi or spectrogram generation from human body movements by studying the temporal and semantics alignment [6, 7, 8, 9, 10]. More recently, diffusion-based methods have been proposed to generate music waveforms directly from videos [11]. In visual-to-natural sound generation, earlier efforts pioneered the generation of sounds linked to various objects and materials [12]. Later works proposed an audio generation approach based on SampleRNN [13, 14] that could generate several types of natural sounds from in-the-wild videos. While these approaches showcase promising results, they are often limited to specific audio categories. Neural codec [15, 16, 17, 18] and autoregressive transformer architectures [19, 20] addressed these limitations and as they have evolved, generative models now effectively generalize across a broader range of sounds or music, leveraging compressed latent spaces [21, 22, 23, 24]. Similar advances have been shown with diffusion techniques such as [25, 26]. However, these methods often lack detailed sound control and their inference time turns out to be consuming. Our work aims to address these limitations by introducing a text-guided framework to improve controllability and efficiency in video-to-audio generation. While there are several concurrent works that aim to achieve partially similar goals to our proposed method [27, 28, 29], our work is different since it is designed to achieve these capabilities within a single unified framework.

### 2.2 Text-to-Audio Generation

As an alternative to the generation of audio from video, text can be used as an input to guide audio generation. When text is the input, audio generation becomes more controllable semantically. Existing approaches such as Make-An-Audio [30], AudioLDM [31], AudioLDM-2 [32] and others [33, 34, 34, 35] enable general text-to-audio (or music) generation by adapting latent diffusion techniques, first developed in [36]. Concurrently, methods such as AudioGen [37], MusicGen [38], AudioLM [39], MusicLM [40], SoundStorm [2], VampNet [41] leverage transformer-based architectures and token-based modeling techniques to produce audio tokens, that are then decoded into waveforms using

neural codecs like Encodec [18] and SoundStream [42]. Notably, SoundStorm and VampNet use an efficient technique known as masked token-based modeling which speeds up generation with parallel unmasking in the decoder. In our work, we consider a similar approach. While these models deliver high-quality audio with strong relevance to the text, they do not necessarily align with visual dynamics when adapted to video-to-audio generation. This is expected since such models have not been trained to attend to visual inputs. Our work addresses this by integrating a pretrained large language model (LLM) as a multi-modal encoder that processes both visual and textual inputs such that the generated audio considers both visual and text information.

## 2.3 Multi-modal Large Language Models

Multi-modal Large Language Models (MLLMs), have been able to attain significant progress. With the advent of open source, pretrained and instruction-tuned LLMs such as LLama [43], Alpaca [44], Vicuna [45]. In particular, when extending these LLMs into MLLMs, a pretrained modality-specific encoder extracts the features and then a projection layer maps these features into vectors of the same dimension as text embeddings of the corresponding LLM. This approach led to developments in visual LLMs [46, 47], audio LLMs [5, 48], audio-visual LLMs [49] and showed improvement in multi-modal understanding tasks such as captioning [50] and question-answering [51, 52]. Recent efforts have also focused on tasks such as multi-modal retrieval [53], multi-modal embodied navigation [54, 55], leveraging LLM's strong reasoning capabilities to interpret or improve the results. In terms of generation, several works [56, 57] aimed at achieving any-to-any modality generation using LLMs as a central medium. While these methods have been successful in general modality-to-modality generation, they do not achieve particular end-to-end video-to-audio generation, with or without text guidance, which is the unique direction our work focuses on.

## 3 Methods

VATT is a flexible vision-to-audio generative framework that can process both visual and textual inputs and generate both audio waveforms and captions of audio. To achieve this, VATT consists of two modeling stages: i) **Video-to-Caption** : This stage utilizes a Large Language Model (LLM) with a learnable projection layer that converts video features into embeddings compatible with the LLM. The model receives an instruction to generate audio captions from video inputs. ii) **Video + Text to Audio**: This stage incorporates an encoder-decoder architecture. The encoder uses the finetuned LLM from Video-to-Caption stage with frozen weights. The decoder is a bi-directional transformer trained to generate audio tokens using masked token modeling techniques in training. The training pipeline of VATT system is shown in Figure 2. During inference, VATT generates audio tokens from video and optional text prompts through iterative parallel decoding. These tokens are then converted into audio waveforms using Encodec [17].

### 3.1 Video-to-Caption Stage

**VATT Converter** is designed to integrate visual and textual prompts for audio generation as well as audio captioning. The core component, *VATT Projector*, is an embedding layer that maps video features into the text embedding space of the LLM. Given visual features extracted from frame-level vision encoders $V_f = \{v_1, v_2, ..., v_T\}$, a Linear layer is applied to project each feature from its original dimension $d_v$ to the LLM text embedding dimension $d_{lm}$, producing a sequence of transformed features $V_{lm} = V_f W_l + b_l$, where $W_l$ and $b_l$ are learnable parameters of the linear projection.

**V2A Instruction Tuning**: The key functionality of VATT Converter is to extract from visual stream semantic features relevant to audio. Drawn on the success of multi-modal LLMs, such as visual-LLM [46] and audio-LLM [5], we employ multi-modal instruction tuning to align the visual inputs of videos with the ground truth audio captioning of the same videos. Given a prompt instruction, $T_i = \{t_{i1}, t_{i2}, ..., t_{iK}\}$, such as "Describe the audio that the video could generate:" and the projected visual features $V_{lm}$ as inputs, we model conditional distribution of audio descriptions $T_a = \{t_{a1}, t_{a2}, ..., t_{aN}\}$, as $P_\theta(T_a|T_i, V_{lm})$ by fine-tuning an instruction-tuned LLM, e.g., Vicuna-7B [45]. Unlike typical instruction-tuning that maps a signal into textual concepts within the same modality, our method bridges the concepts from visual to audio modality, unifying the representation for text-guided video-to-audio generation task that we describe in section 3.2. For training efficiency,

Stage 1: **Video-to-Caption**

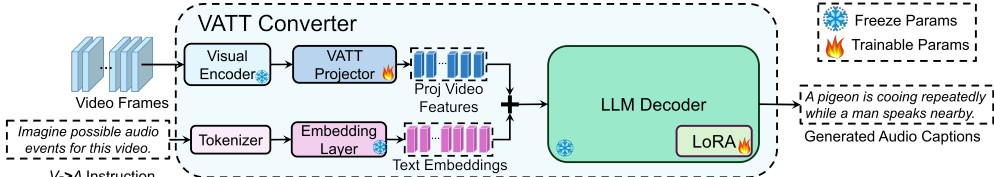

Stage 2: **Video + Text to Audio**

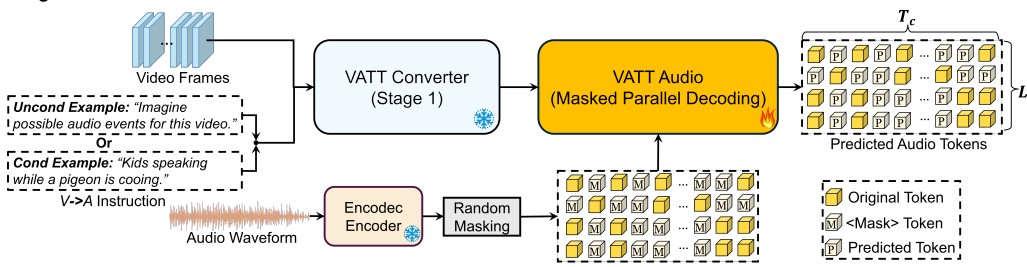

Figure 2: Two stages of *VATT* system training pipeline: (1) **Video-to-Caption** stage that maps video features into an audio caption through LLM. (2) **Video + Text to Audio** stage that learns to generate audio tokens through masked tokens prediction conditioned on Stage (1) features.

we fine tune the LLM with VATT Projector by integrating LoRA [58] adaptors while keeping the original LLM weights frozen. We minimize the negative log-likelihood of audio caption tokens conditioned on visual inputs and prompt instruction

$$\mathcal{L}_{v2t}\left(\widehat{T_a} \mid T_i, V_{lm}\right) = -\sum_{l=1}^{N} log\left[P_\theta\left(\hat{t}_{al} = t_{al} \mid T_i, V_{lm}\right)\right], \tag{1}$$

where $t_{al}$ is the $l$-th text token in the ground truth audio description $T_a$, and $\theta$ is the set of trainable weights including VATT Projector and LoRA adaptor. Further details of the constructions of text prompts and synthesis of audio captions are described in Section 4 and Appendix C.

## 3.2  Video + Text to Audio Stage

Once the audio-related visual features are aligned with the text features in the LLM embedding space, the LLM effectively encodes multi-modal information that serves as a representation for text generation and audio generation. Indeed, in the second stage of VATT, there are two generation modes to generate audio: i) When no conditional text prompt is provided, the video features along with a *standard template* prompt (e.g., "Describe possible audio that the video could infer.") are fed as inputs to VATT Converter. ii) When an audio caption is provided as the text prompt, the video features and the audio caption are fed together into VATT Converter. In such a case, the provided audio caption helps guide the video-to-audio generation process and overrides the need for generated audio caption.

### 3.2.1  Audio Token Decoder

To generate audio, we design an audio token-based decoder, VATT Audio, conditioned on the encoded features from VATT Converter. In contrast to existing methods, which typically use auto-regressive token modeling [37, 38, 23] or latent diffusion techniques [31, 32], we adopt a novel token-based modeling technique based on masking tokens. The method, originally derived in image generation tasks [1] and recently adapted to text-to-audio generation [2, 41], is capable of achieving competitive generation quality while improving efficiency through an iterative parallel decoding algorithm during inference.

**Token-based Representation for Audio** To represent audio waveforms using discrete tokens, we adopt a pretrained audio neural codec, Encodec [17], similarly to FoleyGen [23]. Encodec is a multi-level residual vector-quantized (RVQ) autoencoder trained with waveform reconstruction and

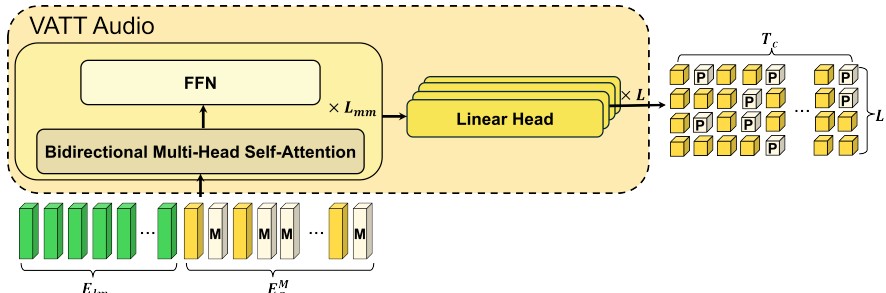

Figure 3: Audio Tokens Decoder: VATT Audio is a bi-directional transformer that models the audio tokens and the conditioning inputs (LLM hidden states) jointly. We extract the part that corresponds to the audio features and apply $L$ Linear layers in parallel to perform classification on masked tokens at each codebook layer.

adversarial objectives, capable of high-fidelity reconstruction from compressed tokens. Specifically, Encodec uses $L = 4$ codebooks of tokens to represent the audio. Lower-level codebooks encode coarse semantic information, while higher-level codebooks capture fine-grained details. We adopt an open source Encodec model pretrained using audio waveforms at $Sr_w = 16kHz$ sampling rate. The model compresses a waveform into tokens at $Sr_t = 50Hz$ sampling rate, leading to $r_{tw} = \frac{Sr_w}{Sr_t} = 320$ waveform samples per token. For any waveform $A_{wav} \in \mathbb{R}^{1 \times T_w}$, we extract corresponding audio tokens representation $A_{tok} \in \mathbb{N}^{L \times T_c}$ ($T_c = \frac{T_w}{r_{tw}}$) from Encodec encoder part. Once the model generates $A_{tok}$, the embedding vectors of $L$ levels of tokens at each time step are summed up before being sent to Encodec decoder to obtain the waveform.

**Masked Audio Token Generative Modeling** We model the distribution of audio token matrix $A_{tok} \in \mathbb{N}^{L \times T_c}$ by developing a token masking strategy which learns the joint distribution of the audio tokens in full parallelism. This is different than using "delayed patterns" proposed in [38] which enables parallelism but only on the level of codebook dimension. At each time step of $A_{tok}$, embedding vectors of $L$ tokens are summed up to represent audio waveform at the corresponding segment. In order to perform masking operation at any position, we introduce an additional learnable <MASK> token in each codebook. By randomly replacing some of the tokens entries in the $A_{tok}$ with <MASK> at corresponding codebook we obtain the masked audio token matrix $A_{tok}^M \in \mathbb{N}^{L \times T_c}$. We obtain $E_a^M \in \mathbb{R}^{d_{em} \times T_c}$ by summation of the embedding vectors of each token in $A_{tok}^M$ along the level axis.

Conditional generative modeling is implemented as follows. We extract the hidden states of the last layer $H_{lm} \in \mathbb{R}^{d_{lm} \times T_{lm}}$ (before the LLM prediction head) from VATT Converter as the conditional inputs into the audio token decoder. We use a linear layer to project $H_{lm}$ to $E_{lm} \in \mathbb{R}^{d_{em} \times T_{lm}}$ with same feature dimension as the masked audio embeddings $E_a^M$. A straightforward way to model the relationship between $E_a^M$ and $E_{lm}$ is to use an interleaving self-attention and cross-attention block as proposed in Vanilla Transformer architecture [59]. However, we find that such interleaved interaction between audio and multi-modal input condition does not capture the fine-grained correspondence between them. Therefore, we propose to use a bi-directional self-attention architecture to fuse the features.

Specifically, we concatenate $E_{lm}$ with $E_a^M$ along the temporal axis to obtain the fused features $E_{mm} = Concat([E_{lm}, E_a^M])$. The decoder consists of $L_{mm}$ layers of self-attention blocks, as shown in Fig. 3. The output hidden states in the last layer of the decoder, $H_{mm}^{out} = Dec(E_{mm})$, represent fused audio and conditions features. We only extract the part of the hidden states corresponding to the audio tokens, $H_a^{out} \in \mathbb{R}^{d_{mm} \times T_c}$, and pass it through $L$ Linear layers in parallel to perform classification on masked tokens at each level of the codebooks. For each masked audio token in matrix $A_{tok}^M$, we calculate the cross-entropy loss between the predicted token $\hat{a}_{tok}$ and the ground truth token $a_{tok}^{gt}$, formulated as

$$\mathcal{L}_{VATT} = - \sum_{a_{tok} \in A_{tok}^M} \mathbb{I}(a_{tok} = \text{<MASK>}) \log \left[ P_\phi(\hat{\mathbf{a}}_{\mathbf{tok}} = a_{tok}^{gt} | A_{tok}^M; H_{lm})) \right], \quad (2)$$

where $\phi$ is the set of trainable parameters in the audio token decoder, and $\mathbb{I}$ is the indicator function.

### 3.2.2 Masking Design and Iterative Parallel Decoding

**Masking Distribution Design** Inspired by [1, 2], we incorporate variable random masking. In particular, it was shown that masking ratio plays an important role in audio token decoder to generate meaningful signals. While in [1, 2] arc-cosine masking distributions is used by default, here we study several masking strategies that include distributions along with different hyper-parameters to find the strategy that reaches more optimal generation quality (see Appendix A for further details). Our study shows that normal distribution with a mean of 0.75 and standard deviation of 0.25, truncated from 0.5 to 1.0 is such optimal strategy. The general interpretation of this strategy is that a relatively high range masking ratio enables models to generate better initial tokens when most of the entries in the token matrix are masked. This is essential for future decoding steps to generate meaningful tokens.

**Iterative Parallel Decoding** Scheduling of masking plays a key role as well. During inference, we follow the cosine scheduling scheme proposed in [1] to gradually resolve the audio tokens. The iterative sampling procedure starts with all <MASK> in the audio token matrix. At a step $t$, the model takes the audio token matrix $A_{t-1}$ from the previous step along with the conditions as inputs and samples a new audio token matrix $\hat{A}_t$ in parallel with all tokens unmasked. Based on the confidence at each entry of $\hat{A}_t$ only tokens with top-k confidence are kept while the remaining entries are re-filled with <MASK>, resulting in $A_t$. The cosine scheduling scheme determines the ratio of re-masked tokens by $r_t = cos\left(\frac{\pi}{2} \cdot \frac{t}{T}\right)$. Notably, to resolve the confidence of each entry in the matrix, we adopt the *"gumbel-top-k trick"* [60] with temperature that varies, i.e., $c_i = \frac{log(p_i)}{\tau} + G$, where $G \sim \text{Gumbel}(0, 1)$ and $p_i$ denotes the output probability of the sampled token at the entry $i$. This is equivalent to sampling k values from multinomial distribution from the softmax probabilities without replacement. The temperature $\tau$ controls the degree of stochasticity. We use $\tau = \tau_0 \cdot (1 - \frac{t}{T})$ with linear decay during generation, where $\tau_0$ is the initial temperature. Similarly to [1, 2], our method achieves optimal quality and fast speed within a few decoding steps (typically 10 - 20).

## 4 Experiments

**Datasets:** We use common benchmarks datasets VGGSound [3] and AudioSet-2M [4] for training and evaluation. VGGSound is a large-scale audio-visual dataset sourced from YouTube, containing 192k videos from 309 audio-visual categories, with 177k / 15k train-test video splits. AudioSet-2M is a larger audio-visual database with around 2M YouTube videos, with only 1.6M available online. In Stage 1, we train VATT Converter with both datasets and test on VGGSound only. In Stage 2, for fair comparison against existing video-to-audio generation methods, we train and evaluate on VGGSound dataset only.

To train VATT with text, we synthesize a large-scale audio caption dataset, "V2A Instruction", using LTU [5], an existing audio LLM. We obtain audio captions by prompting the pretrained LTU-13B model with the inputs of audio waveform along with the instruction *"### Instruction: Close-ended question: Write an audio caption describing the sound. ### Response:"*. For AudioSet [4] and VGGSound [3] we generate a single audio caption per each video for a total of 1.77M videos.

To ensure the quality of captions, we first manually verified the validity of LTU-generated captions prior to using them as synthetic ground-truth (GT) and then performed an experiment to further evaluate captions quality. In particular, we randomly selected 100 videos from VGGSound test set with stratified sampling according to video categories to conduct a human study. We used 1-5 point MOS (Mean-Opinion-Score) scale (the higher the better) to measure correctness of the captions. We provide pairs of videos and the corresponding captions to the raters, asking "How accurately the provided caption reflects the sound events happening in the video? 1. Inaccurate and irrelevant. 2. Relevant but inaccurate with many mistakes. 3. Partially accurate but missing details and with mistakes. 4. Mostly accurate with some minor mistakes. 5. Accurate and complete." We used the MTurk platform to perform the evaluation and collected a total of 300 responses. The generated captions have a high MOS of mean 4.72 and std 0.37, providing an additional indication for the validity of the synthetic ground truth.

**Implementation Details:** For visual inputs, we use eva-CLIP [61] image encoder to extract mean-pooled visual features from video frames at 5fps rate, which result in $50 \times 768$ visual sequence for a 10s video. To represent audio, we extract audio tokens from a pretrained Encodec-16kHz. For each 10s audio waveform, we represent it with $A_{tok} \in \mathbb{N}^{4 \times 500}$ token matrix.

Table 1: Quantitative results against video-to-audio generation methods on VGGSound test set. '-T' refers to model with text prompts.

| Methods | KLD ↓ | FAD ↓ | Align Acc ↑ | Speed (s) ↓ |
|---|---|---|---|---|
| SpecVQGAN [21] | 3.78 | 6.63 | 48.79 | 7.2 |
| IM2WAV [22] | 2.54 | 6.32 | 74.31 | 289.5 |
| Diff-Foley [25] | 3.15 | 6.40 | 82.47 | 4.4 |
| FoleyGen [23] | 2.89 | 2.59 | 73.83 | 6.9 |
| V2A-Mapper [26] | 2.78 | **0.99** | 74.37 | 11.54 |
| **VATT-LLama (Ours)** | 2.39 | 2.38 | 80.32 | **1.1** |
| **VATT-Gemma (Ours)** | **2.25** | 2.35 | **82.81** | **0.65** |
| **VATT-LLama-T (Ours)** | **1.41** | 2.54 | 80.16 | **1.2** |
| **VATT-Gemma-T (Ours)** | **1.66** | 2.98 | 81.48 | **0.76** |

For LLM, we explore two open-source models, Gemma-2B [62] and LLama-2-7B [43], using instruction-tuned checkpoints. The LLM hidden size of Gemma-2B is 2048 and 4096 for LLama-7B. For both LLMs, we train VATT Converter using LoRA parameter-efficient fine-tuning technique while keeping the LLM weights frozen. We use rank $r = 16$ and $\alpha = 32$ with 0.1 dropout rate for LoRA configuration.

VATT Audio is a bi-directional transformer with 24 layers, each with hidden size 1024 with 16 attention heads. To differentiate the conditioning inputs and audio tokens, we add two learnable modality-specific embeddings with respect to the corresponding inputs(see further implementation details in AppendixD).

**Evaluation Metrics:** To evaluate video-to-audio generation quality, we follow the method of [23], which proposed the metrics Kullback-Leibler-Divergence (KLD) with PassT [63], Fréchet Audio Distance (FAD) [64] and Align Accuracy (Align Acc) [25]. KLD measures how closely the generated audio matches the GT through pairwise comparison, reflecting how well the audio captures the concepts in the video. FAD evaluates the overall distribution, indicating the overall quality of the audio. Align Acc assesses the relevance and temporal alignment of the audio and the video. Additionally, we incorporate generation speed (time taken per waveform sample) to measure efficiency. We also compute the CLAP score [65] to evaluate the adherence of generated audio to text prompts to compare our results with text-to-audio generation. Further details of these metrics are described in Appendix F.

For video-to-audio captioning, we use two types of metrics, natural language generation (NLG) metrics and audio-text relevance metric. NLG metrics evaluate the generated captions with respect to the ground truth audio captions using rule-based matching in terms of precision and recall. These metrics include BertScore [66], BLEU-4 [67], ROUGE-L [68] and CIDEr [69]. To assess the relevance of generated audio captions with the actual audio, we compute the CLAP-score [65] as cosine similarity between audio and text embeddings.

**Quantitative Evaluation of Audio Generation:** We evaluate audio generation of VATT models on the VGGSound test split. For each of the 15,446 video samples, we generate a 10-second audio waveform. We compare VATT variants against existing video-to-audio generation methods as well as text-to-audio generation methods including AudioLDM-2 [32] and AudioGen [37] using different text prompts. The results on the metrics described above are summarized in Table 1 and Table 2. VATT models achieve best KLD score and Align Acc against other methods while maintaining competitive FAD (top 2). Notably, when guided by GT audio captions (VATT-LLama-T and VATT-Gemma-T; bottom) our models generate sounds that match the GT audio more accurately, as indicated by lowest KLD score of 1.41 and 1.66 for VATT models with two LLM backbones, surpassing both video-to-audio and text-to-audio methods. In comparison to text-to-audio methods, VATT models achieve competitive audio-text alignment in terms of CLAP score, demonstrating a strong capability to follow text prompts. Implementation details of these baselines are included in Appendix E.

**Quantitative Evaluation of Video-to-Audio Captioning:** We evaluate video-to-audio captioning by prompting VATT Converter to generate audio captions. We use the prompt "Describe the possible audio for this video:" to generate captions for all VGGSound test videos. For baselines, we prompt LLAVA-13B-v1.5 model in two zero-shot modes to generate visual and audio descriptions respectively. Since LLAVA can take a single image as an input only, we select the middle frame of videos. We

Table 2: Quantitative results comparing VATT with text-to-audio generation methods on VGGSound test set. '-T' refers to model with text prompts. CLAP score is calculated as the cosine similarity of generated audio with respect to the GT audio caption.

| Methods | Text Prompt | KLD ↓ | FAD ↓ | Align Acc ↑ | CLAP Score ↑ |
|---|---|---|---|---|---|
| AudioGen [37] | LLAVA visual caption | 3.65 | 6.03 | 41.66 | - |
| AudioGen [37] | GT audio caption | 2.19 | 3.17 | 48.96 | **0.409** |
| AudioLDM-2 [32] | LLAVA visual caption | 3.54 | 3.62 | 53.49 | - |
| AudioLDM-2 [32] | GT audio caption | 2.09 | **2.46** | 51.84 | 0.326 |
| **VATT-LLama-T (Ours)** | GT audio caption | **1.41** | 2.54 | **80.16** | 0.347 |
| **VATT-Gemma-T (Ours)** | GT audio caption | **1.66** | 2.98 | **81.48** | 0.310 |

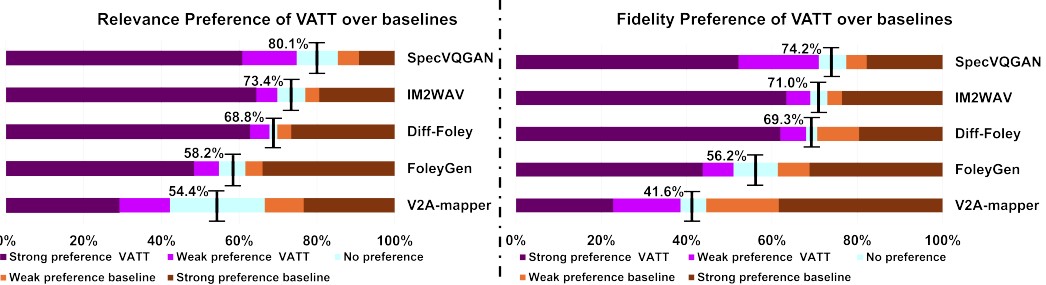

Figure 4: Qualitative evaluation results: Pairwise Comparison of generated audio VATT v.s other methods comparing Fidelity and Relevance aspects.

use "Provide a concise, descriptive caption for the following image." as the visual prompt, and "Describe the sounds that this scene could yield in a short sentence without reasoning" as the audio prompt. We also compare against a video LLM baseline, Video-LLAMA-7B, to perform zero-shot video-to-audio captioning. Specifically, we directly input VGGSound videos into the VL branch of the Video-LLAMA model, and prompt it to generate audio captions using the instruction "User/ What sounds could match the video?" Since Video-LLAMA has not been pretrained on VGGSound dataset and LTU generated captions, we implement a similar structure of Video-LLAMA and train on our LTU-generated captioning data. We replaced the original BLIP-2 visual features used by Video-LLAMA with our eva02-CLIP-L visual features due to the expensive pre-processing time for all BLIP-2 features from videos in VGGSound and AudioSet. For the Video-QFormer component of Video-LLAMA, we keep it the same as Video-LLAMA, and we name this model as VATT-Qformer - LLama. Our evaluation is summarized in Table 3. VATT models with LLMs outperform LLAVA-prompted and Video-LLAMA zero-shot results demonstrating a stronger capability to infer sounds from videos semantically. In particular, when measuring audio-text relevance, our model with LLama achieves an increase of **+5.0%** in accuracy when compared with LLAVA visual caption baselines. For reference, the ground truth audio captions generated by LTU [5] have an average CLAP score of 0.379.

Table 3: Comparison of video-to-audio captions on NLG evaluation metrics and text-audio relevance (CLAP Score).

| Methods | BertScore (F1) ↑ | BLEU-4 ↑ | ROUGE-L ↑ | CIDEr ↑ | CLAP Score ↑ |
|---|---|---|---|---|---|
| LLAVA w/ Visual Prompt | 0.855 | 0.089 | 0.137 | 0.026 | 0.213 |
| LLAVA w/ Audio Prompt | 0.870 | 0.123 | 0.155 | 0.095 | 0.182 |
| Video-LLAMA w/ Audio Prompt | 0.861 | 0.091 | 0.117 | 0.021 | 0.204 |
| VATT Converter - Gemma (ours) | 0.900 | 0.345 | 0.337 | 0.926 | 0.229 |
| VATT-Qformer - LLama | 0.907 | 0.419 | 0.375 | 1.264 | 0.245 |
| VATT Converter - LLama (ours) | **0.909** | **0.424** | **0.384** | **1.354** | **0.263** |

**Qualitative Evaluation:** In addition to quantitative evaluations, we also conduct a qualitative (subjective) study to evaluate audio generation perceptual quality of VATT. Specifically, we randomly select 100 videos from VGGSound test split with stratified sampling according to video categories. For each method in the baseline, we pair the generated samples against VATT. Two aspects of the generation are evaluated, Fidelity and Relevance. Fidelity focuses solely on audio quality, while Relevance evaluates the semantic relevance and temporal alignment of audio to the video. For each

Table 4: Architecture Ablation Study.

| Methods | KLD ↓ | FAD ↓ | Align Acc ↑ |
|---|---|---|---|
| VATT-V | 2.43 | 2.53 | 82.43 |
| VATT-Cross-Attn | 2.76 | 3.63 | 76.85 |
| **VATT-Gemma (Ours)** | **2.25** | **2.35** | **82.81** |

pair, raters are asked to rate their scoring of VATT versus a compared baseline on a Likert scale from 1 (strongly prefer baseline) to 5 (strongly prefer VATT). We use our best VATT variant, VATT-LLama-T (with GT text guidance), for the comparison. As shown in Table 4, VATT surpasses other methods in Relevance. In terms of Fidelity, VATT is consistently being preferred when compared with most baselines and slightly less preferred when compared with V2A-Mapper. The reason could be that V2A-Mapper is directly optimized with diffusion techniques on AudioLDM, a large-scale pretrained text-to-audio model, such models tend to perform better in fidelity aspect in comparison to token-based models. Further details of qualitative evaluations are incorporated in Appendix G, and qualitative samples are provided in Appendix B.

**Ablation Studies:** We study the effectiveness of VATT Converter by removing the LLM and directly feed the visual features into the decoder. We denote such model as VATT-V. While VATT-V does not handle textual inputs or generate text, it still serves as a strong variant of VATT for video-to-audio generation. To study the contribution of audio token decoder, we replace the decoder part of VATT with interleaving attention blocks proposed in vanilla transformer [19], and denote this variant as VATT-Cross-Attn. As shown in Table 4 VATT-Gemma model outperforms both VATT-V and VATT-Cross-Attn. When VATT is conditioned on visual inputs only its performance is lowest across variants.. The VATT Converter enhances the visual features through audio-relevant text, thereby improving the relevance and quality of the generated audio. In addition, we find that the bi-directional transformer design in VATT Audio is critical for learning the associations between audio and conditioning inputs to enhance audio generation performance. Additional ablation studies can be found in Appendix A.

## 5 Conclusion

In this work, we propose a multi-modal generative framework that enables both text-guided video-to-audio generation and video-to-audio captioning. Experiments show that our method can generate high quality audio through text in both unconditional and conditional modes, as well as to generate reasonable audio captions from videos. One area for improvement is the diversity of the text generated by current audio LLMs. In cases where the user-provided text prompts significantly differ in style there is a possibility for a conflict of audio quality and adherence to the instructions. Future work could enhance the capability of the model to generalize across different text styles and to further develop capabilities for informative iterative conversation-like video-to-audio generation.

## Broader Impact

VATT could augment existing audio-video creation tools for content creators by allowing generation of custom audio tracks for given visual content through user provided text prompts. Also, VATT has the ability to suggest potential sounds for a given video which can inspire creators by presenting audio options that may not have been considered otherwise. This feature can be useful for brainstorming of content creation, where audio choices can influence the style of the final product.

Further extensions of this work could involve conversational video-to-audio generation such that the audio content is iteratively being refined. By integrating a conversational interface, the users can engage in a dialogue with the system, making requests and receiving responses. This approach goes beyond static text inputs, offering a more accessible toolset that does nto require significant audio editing expertise. Moreover, the conversational system can seek clarifications or propose alternatives, functioning like an assistant to avoid misunderstandings and enhance audio quality. More broadly, the generative approach proposed here has the potential to adapt to other generative areas not limited to audio, video, but also potentially impact fields such as biochemistry, physics where a generative approach is utilized, e.g., generative modeling of high-energy particle events [70].

While VATT presents a potential for content creation, the ability to generate realistic audio from visual inputs could lead to misuse, such as creating deceptive content or deepfake audio and ethical concerns must be addressed before utilization. Furthermore, similarly to audio generation, text generation capability could result in misuse such as offensive language or privacy violations. To mitigate these risks, in further development or potential code release we will establish clear ethical guidelines, evaluate for biases, and implement safeguards to ensure responsible use and fair outputs.

## Acknowledgments and Disclosure of Funding

We acknowledge the support of HDR Institute: Accelerated AI Algorithms for Data-Driven Discovery (A3D3) National Science Foundation grant PHY-2117997 and the departments of Applied Mathematics and Electrical and Computer Engineering at the University of Washington.

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

# Appendix

In this Appendix, we provide:

- Additional Ablation Studies and Comparisons, see Appendix A.

- Qualitative Examples and Analysis, see Appendix B.

- Details and Examples of our synthetic "V2A Instruction" Dataset, see Appendix C.

- Additional Implementation Details of VATT, see Appendix D.

- Additional Implementation Details of baselines, see Appendix E.

- Details of Evaluation Metrics, see Appendix F

- Human Evaluation Details, see Appendix G

## A  Additional Ablation Studies and Comparisons

**Masking Ratio Distribution Ablation:** Designing an appropriate varying masking ratio distribution for training is essential to achieve high audio quality and relevance. We study several commonly used masking ratio distributions, including Uniform distribution, Gaussian distribution and arc cosine distribution. For Gaussian distribution, we experiment with distributions with 4 different mean values, 0.55, 0.75, 0.95, and moving mean following a sine schedule with respect to the training epoch (similar to curriculum learning), in the range of $[0.25, 0.95]$. The standard deviation is kept fixed at 0.25. We use VATT Gemma-2B model for the masking ratio ablation study. As shown in Table 5, the model performs better when the distribution has a higher mean in masking ratio, especially arc cosine distribution and Gaussian mean with 0.75. This is due to the fact that the initial steps during the sampling stage are important for future decoding steps. The initial steps correspond to high masking ratio cases. For later steps, new tokens are unmasked conditioned on more clues such that the masking ratio decreases and the generation becomes less challenging, thus making the learning at lower masking ratio easier during training.

**Self-prompting Ablation:** When additional text prompts are provided as inputs, we could use the audio captions generated by the VATT Converter as the text prompt to generate the audio. In this self-prompting mode, the generated audio could be interpreted by the same model in terms of the caption. As shown in Table 6, when our model is fed with corresponding generated captions, the model performs slightly worse than the model without prompt input, showing the space for improvement in the quality of generated captions. Also, generation of audio with the captions from VATT-Converter-LLama outperforms captions from VATT-Converter-Gemma, in particular evident from the FAD and KLD scores. As the caption quality improves, the text-conditioned video-to-audio generation performance also improves. The GT audio captions generated by LTU obtains the highest CLAP score (measured with respect to the GT audio in original video) of 0.379, reflecting the best caption quality. Feeding such GT captions as input to the model also leads to the best audio generation results.

**Comparison with Text-to-Audio generation methods on AudioCaps:** We use VGGSound as our main dataset and benchmark to evaluate since it is a large-scale audio-visual dataset with around 200K videos across many categories, and also the quality of audio-visual alignment is high. To further show the generalization capability of VATT, we experiment with AudioCaps dataset. Due to limited video samples in AudioCaps, we finetuned our VGGSound pretrained VATT model on AudioCaps dataset in two settings, with and without text prompts. To keep the comparison fair, we use the GT audio captions from AudioCaps as the text prompts. We use VATT-LLama and VATT-LLama-T to compare against AudioGen and AudioLDM-2. As shown in Table 7, VATT-LLama-T performs on a similar level to AudioGen in terms of FAD and KLD score, while falling behind AudioLDM-2. It is noteworthy that the audio decoder of both AudioGen and AudioLDM-2 are pretrained on much larger data scale (7000 hrs and 30000 hrs audio respectively) than ours (700 hrs audio). Despite this, VATT still performs reasonably well on this dataset.

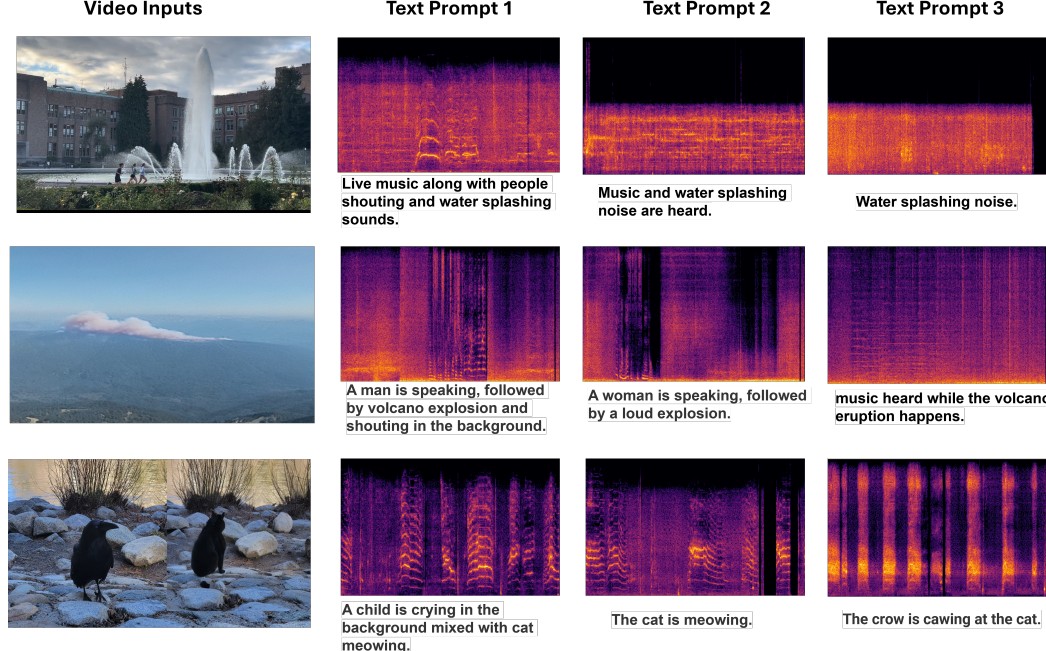

Figure 5: Qualitative samples that showcase text controllability: For same video inputs, VATT is able to generate different sounds that align with the additional text prompts, showcasing its capability of performing controllable generation.

# B Qualitative Examples and Analysis

**Controllable audio generation through text prompt:** A unique advantage of our model lies in its capability to control the generated details of audio through text prompts. We show a few samples where different text prompts are applied to the same video to generate different variations of sounds. As shown in Figure 5, our model is able to generate reasonable sounds that are distinct in their semantic meaning but fit both the context of the video and are aligned with the text description. The text prompts shown are all human-written prompts rather than synthetic ones.

**VATT without prompt v.s VATT with ground truth audio captions:** We also compare generation using our model with GT audio captions as prompt versus generation without prompt to understand why the KLD score with prompt outperforms the generation without prompt by a large margin. Upon inspection of generated samples, as shown in Figure 6, we find that GT audio captions could steer the model generation towards the GT audio in the test set. For example, the first video shows a man performing with a rope, so the rope tapping sounds (when hitting the ground) should be heard in the video. However, it occurs that the model without prompt fails to capture this important detail, but instead generates noises from the surrounding crowd. Similar cases apply to the other two examples shown in the figure. KLD measures the pairwise difference between generated sounds and GT sounds in the feature space. Therefore, a low score means that the model closely matches the semantic meaning of the GT audio in the original video, which indicates that the model is able to follow the text prompt instruction to generate the desired audio.

**Video-to-Audio Captioning:** In addition to controllable video-to-audio generation through text, VATT is also able to generate audio captions from videos, providing textual suggestions interpreting what sounds could a given video make. As shown in Figure 7, VATT could produce reasonable audio captions for videos across a variety of audio-visual categories, showcasing the capability of VATT Converter in capturing the audio relevant features from the video.

Table 5: Ablation on Masking Ratio Distribution.

| Methods | KLD ↓ | FAD ↓ | Align Acc ↑ |
|---------|-------|-------|-------------|
| Uniform | 2.52 | 2.75 | 80.37 |
| Arc cosine | 2.34 | 2.26 | 82.88 |
| Gaussian w./ mean 0.55 | 2.31 | 2.34 | 82.27 |
| Gaussian w./ mean 0.75 | 2.25 | 2.36 | 82.81 |
| Gaussian w./ mean 0.95 | 2.24 | 2.49 | 81.80 |
| Gaussian w./ moving mean | 2.42 | 2.32 | 81.81 |

Table 6: Ablation on self-prompting text-guided generation.

| Methods | Text Prompt | KLD ↓ | FAD ↓ | Align Acc ↑ |
|---------|-------------|-------|-------|-------------|
| **VATT-LLama** | ✗ | 2.39 | 2.38 | 80.32 |
| **VATT-Gemma** | ✗ | 2.25 | 2.35 | 82.81 |
| **VATT-LLama-T** | VATT-Converter-LLama | 2.38 | 2.58 | 80.41 |
| **VATT-LLama-T** | VATT-Converter Gemma | 2.57 | 2.67 | 79.20 |
| **VATT-Gemma-T** | VATT-Converter-Gemma | 2.40 | 3.67 | 80.07 |
| **VATT-Gemma-T** | VATT-Converter-LLama | 2.26 | 3.20 | 80.42 |
| **VATT-LLama-T** | GT audio caption | 1.41 | 2.54 | 80.16 |
| **VATT-Gemma-T** | GT audio caption | 1.66 | 2.98 | 81.48 |

Table 7: Quantitative results against text-to-audio generation methods on AudioCaps test set.

| Methods | KLD ↓ | FAD ↓ | Align Acc ↑ | CLAP Score ↑ |
|---------|-------|-------|-------------|--------------|
| AudioGen | 2.09 | 3.13 | 58.26 | 0.447 |
| AudioLDM-2 | 1.64 | 1.86 | 60.32 | 0.432 |
| VATT-LLama | 2.53 | 3.42 | 75.76 | - |
| VATT-LLama-T | 2.07 | 3.25 | 74.89 | 0.376 |

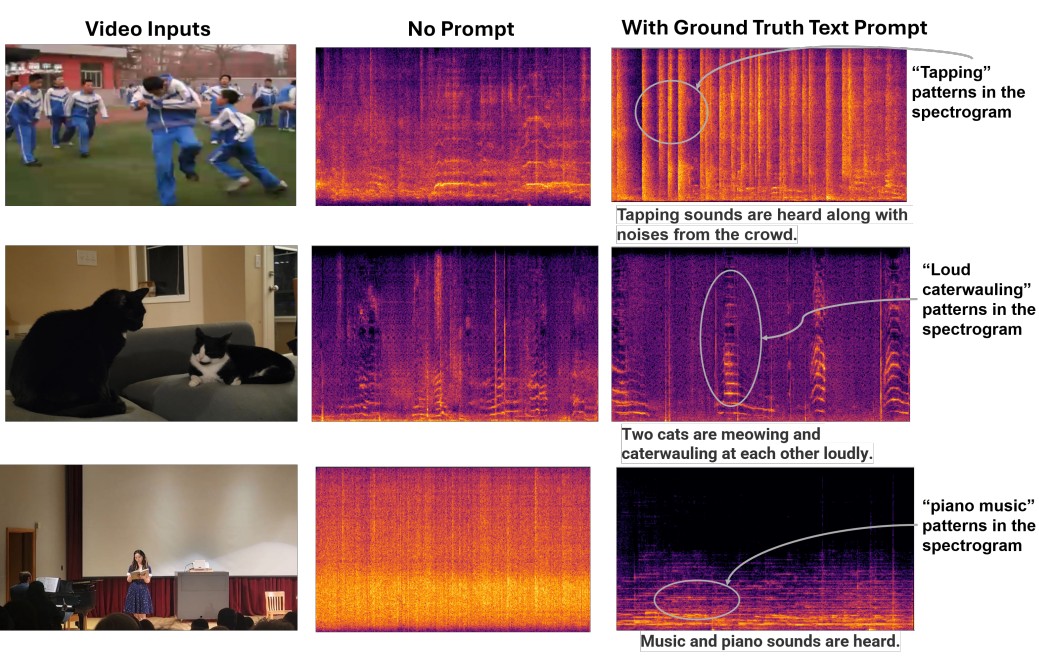

Figure 6: Steering generation towards ground truth audio: For same video inputs, we compare our generation results without text prompt v.s feeding ground truth audio caption as additional prompt.

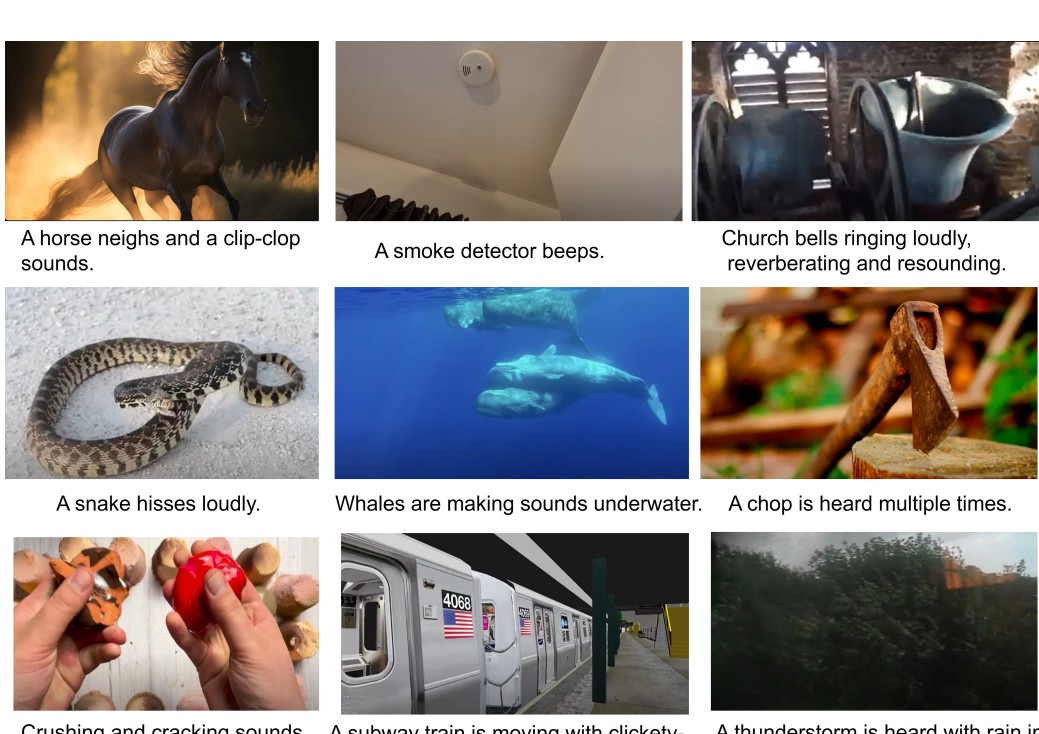

| | | |
|---|---|---|
| A horse neighs and a clip-clop sounds. | A smoke detector beeps. | Church bells ringing loudly, reverberating and resounding. |
| A snake hisses loudly. | Whales are making sounds underwater. | A chop is heard multiple times. |
| Crushing and cracking sounds occur. | A subway train is moving with clickety-clack sounds and background noise. | A thunderstorm is heard with rain in the background. |

Figure 7: Video-to-Audio Captioning samples by VATT.

## C  Details and Examples of V2A Instruction Dataset

We describe the synthesis procedure of our V2A Instruction dataset. To obtain audio captions for training and evaluating VATT, we use existing an existing audio large language model, LTU [5], which is pretrained on a large-scale audio understanding OpenAQA dataset, including audio from VGGSound and AudioSet-2M. LTU demonstrates strong capability in audio captioning in a zero-shot prompting manner, accurately reflecting what happens in the audio.

Specifically, we adopt the prompt "Close-ended question: Write an audio caption describing the sound." used during LTU training for audio captioning task, and feed 10-second audio from VG-GSound and AudioSet dataset as inputs into LTU-13B model (max length 108 version). In figure 8, we show 15 examples of synthesized captions from videos in VGGSound along with the corresponding video IDs and start-to-end time. The generated captions are clear natural language and faithfully describe the audio content in details, serving as a reliable dataset for training and evaluation.

## D  Additional Implementation Details of VATT

**Data Preprocessing:** For visual inputs, we extract video frames at 5fps rate, resulting in 50 frames for each 10s video. For audio, we extract tokens from audio waveform using pretrained Encodec-16kHz, resulting in a $4 \times 500$ token matrix for each 10s audio. To extract visual features, we resize each video frame to $336 \times 336$ and normalize, and feed into the eva-CLIP-L [61] image encoder. By extracting the mean-pooled vector from the hidden states, we represent each video frame with a 768-dim vector. For textual inputs, we use template text prompts as input to instruct video-to-audio captioning, including 10 human-written prompts. For training the VATT Audio, two cases are considered. In unconditional generation case where no additional audio caption is provided, the video along with the one of the template text prompts (shown in Table 8) are fed as inputs to the model. In conditional generation case where the ground truth audio caption is provided, the caption replaces the template prompt as the textual input to the model. For both cases, the textual inputs are formatted using "alpaca short" instruction style, "### Instruction: instruction ### Response:".

| YouTube ID | Start - end time (s) | Audio caption |
|---|---|---|
| r5bG4ZdOZ6M | 1129 - 1139 | Paper is being crumpled and torn with mechanisms making noise. |
| _R6pY_ivHxQ | 70 - 80 | A car is skidding and a man speaks in the background, with tire squealing and speech. |
| eLxrJtNB980 | 190 - 200 | Music is playing and a saxophone is being played in the background. |
| qHgTFW_NMRs | 31 - 41 | Insects are chirping, a man is speaking and roaring lions can be heard. |
| p7DZMOC0ty8 | 30 - 40 | Rain falls and thunder booms. |
| fFTFyawoWiE | 67 - 77 | A steam whistle blows and a train moves with hissing sounds. |
| -CfhAXuBCJQ | 118 - 128 | A man is speaking, basketball bounces and people are clapping, cheering, and whistling while a crowd is heard in the background, with occasional male speech and background noise. |
| _Su1oPvlnL8 | 33 - 43 | Bouncing sounds of a ping pong game are heard. |
| XyjsrZegPnl | 20 - 30 | People are speaking, shouting, and clapping amidst firecrackers, hubbub, and speech. |
| d1yLvXtucLs | 30 - 40 | A church bell rings and ticks. |
| Bl_FyZ4c_pl | 53 - 63 | A beatboxer is performing, with a crowd cheering and speaking sounds. |
| 8qy0Mt_Z8GQ | 30 - 40 | A man is speaking, chopping food, and breathing with background noise. |
| N1VHutvx7jU | 109 - 119 | A choir sings with yodeling and breathing in the background. |
| -JUBdOr8Hes | 30 - 40 | An accordion is playing music, with a reedy and airy quality. |
| C6xMq9aur5Y | 121 - 131 | Wind blows and a person skiing on snow with wind in the background. |

Figure 8: Examples of V2A Instruction Dataset.

| No. | Text Prompt Template |
|---|---|
| 1 | Imagine possible sounds for this video. |
| 2 | Describe possible sounds that could match the video. |
| 3 | What audio could be inferred from this video? |
| 4 | What sounds could match the video? |
| 5 | Infer the sounds that match the video. |
| 6 | What audio could best match this video? |
| 7 | What sound events could the video yield? |
| 8 | Caption possible sound events that describe the video. |
| 9 | What sound events make sense to this video? |
| 10 | Imagine audio events that match this video. |

Table 8: Text prompt templates used in video-to-audio captioning and unconditional video-to-audio generation.

**Training:** For VATT Converter, we perform V2A Instruction-Tuning with our 1.77M audio captions from VGGSound and Audioset altogether. To project visual features to LLM embedding dimensions, we first apply adaptive pooling along the temporal axis, reducing the temporal dimension from 50 to 10 to conserve GPU memory. Then a linear layer is applied on each time step of the visual features to project 768-dim to the LLM dimension (4096 for LLama and 2048 for Gemma-2B). The training process involves two sub-stages. In the first stage, we turn off the LoRA setting and tune the VATT Projector layer only. After 2 epochs, we start the second stage training by tuning both projection weights and LoRA parameters for another 4 epochs. We use AdamW optimizer with a base learning rate of 1e-4, and limit the maximum length of the audio captions to be 108. For both LLama-7B and Gemma-2B, we initialize our model with instruction fine-tuned checkpoints available on huggingface.

For VATT Audio, we explore various masking ratio distribution and end up using a truncated Gaussian distribution with a mean of 0.75 and standard deviation of 0.25, truncated between 0.5 and 1.0. To enable classifier-free guidance during sampling, we randomly replace the conditioning features with vectors of all zeros with 10% rate. The training also takes two sub-stages: i) We first train the model in unconditional generation mode without ground truth audio caption text prompts as inputs until convergence, and then ii) train the model conditioned on GT audio captions as textual inputs (limiting

maximum text length to 64). Additionally, we apply two types of data augmentations during the unconditional training phase to facilitate the temporal alignment between audio and video: i) temporal mixup following [25] with a probability of 0.5, ii) temporal rolling on the video features together with audio tokens by same amount of time. We train our model with a base learning rate of 2e-4 with a warmup step of 40k and linear decay schedule with 600k steps. The number of trainable parameters for VATT Audio is 415M. We use a batch size of 48 for Gemma model and 36 for LLama model.

All our training procedure are conducted on a single A100 80GB GPU, VATT Converter training takes 16 hours and VATT Audio takes 3 days in total.

**Inference:** For Video-to-Audio Captioning, we adopt the generation configuration with temperature of 0.1, top_p of 0.95, top_k of 500 and repetition penalty of 1.1 for both LLMs. We use one of the templates "Describe possible sounds that could match the video." to prompt LLMs to generate captions.

For audio generation, we adopt the iterative parallel decoding strategy with total decoding steps of 16, softmax sampling temperate 1.0 with top_k being 256. For masking sampling with Gumbel top-k strategy, we use initial temperature of 27.5 with linear decay each iteration, where the re-masking ratio follows the cosine schedule, same as [2, 1]. Following [23], we also use classifier-free guidance [71] during sampling, and we find a cfg_scale of 5.0 works best for our model.

# E   Implementation Details of video-to-audio generation baselines

**SpecVQGAN [21]:** We follow the open source SpecVQGAN's codebase instructions to extract visual frame-level RGB features at 21.5fps rate using ResNet-50 checkpoints for all VGGSound test videos. Following the evaluation script setup for VGGSound, we generate a 10-second audio waveform per each video in the test set.

**IM2WAV [22]:** Following the open source codebase of IM2WAV, we extract CLIP visual features from videos at 30fps rate for VGGSound test videos. IM2WAV was originally trained to generate 4-second audio. In order to adapt it to generate a 10-second audio, we infer the model with 3 forward passes to obtain three 4-second audio segments without overlap, and concatenate them together and then trim to a 10-second audio.

**Diff-Foley [25]:** Diff-Foley uses their pretrained CAVP audio-visual model to extract features at 4fps rate. Following their open source codebase, we extract visual features for test videos in VGGSound, and apply their best generation configuration with double guidance scale CFG scale $\omega = 4.5$, CG scale $\gamma = 50$, to generate three non-overlapping 4-second audio segments in the same way as IM2WAV.

**FoleyGen [23]:** The authors do not open source their implementation. We strictly follow the setup in the paper and uses the open source version of Encodec-16kHz to replicate their model. For visual features, we follow their implementation to extract the CLIP features at 1fps rate. FoleyGen is a 24-layers transformer architecture with hidden size of 1024 and 16 heads. Using their best visual attention configuration "All-frame" attention along with random visual condition dropout with a probability of 0.1, we train FoleyGen with specified hyperparameter settings as described in the paper. In the inference stage, we apply classifier-free guidance scale of 3.0 as well as top-k 256 sampling configuration in the paper to generate 10-second video per test video in VGGSound. Upon evaluation, we find that there is a noticeable gap between our implementation results (KLD: 2.89, FAD: 2.59) and reported results in paper (KLD: 2.35, FAD: 1.65). To study where the gap comes from, we use our extracted Encodec tokens to reconstruct the audio waveform in VGGSound test set, and measure the FAD score of reconstructed audio waveform with respect to the ground truth audio. We find that the open source Encodec-16kHz on huggingface could only achieve a FAD score of 1.86, which still falls behind their reported result of 1.65, indicating that the released Encodec model is a sub-optimal version.

**V2A-Mapper [26]:** V2A-Mapper is not yet open sourced, but the authors publish their generated audio for 15,446 video samples in VGGSound test set. We therefore download their samples and conduct both objective and subjective experiments based on them.

# F    Details of Evaluation Metrics

**KLD score**: To compute the KL-Divergence between generated samples and ground truth audio, we adopt the pretrained PaSST [63], an audio transformer classifier, on AudioSet-2M dataset to extract the classification output probabilities. KLD is evaluate in a pairwise manner and we use the mean value over all 15,446 samples as our KLD score.

**FAD score**: Fréchet Audio Distance [64] (FAD) evaluates the difference between the distributions of generated samples and ground truth samples. Specifically, we adopt the pretrained VGG-ish network on AudioSet-2M dataset to extract the features of generated audio and ground truth audio. Using the multivariate gaussian assumption on extracted features, we compute the mean and covariance of the generation sample set and ground truth audio set, and then apply the Fréchet distance formula to obtain the FAD score. Specifically, to ensure the correctness of computation, we use the original implementation of Google Research's tensorflow version to perform evaluation.

**Align Acc:** To evaluate temporal alignment and relevance of audio to video, we also incorporate Align Acc metric proposed by [25]. Specifically, Align Acc is computed using a CAVP (contrastive audio-visual pretraining) model by taking video frames along with the audio mel-spectrogram as inputs. The model outputs an accuracy score representing the alignment between audio and video. Following their configuration, we use visual frames at 4 fps rate as visual inputs (40 frames for 10-second video), and convert the 16kHz audio waveform into mel-spectrogram with FFT Num 1024, mel basis Num 128 and hop size 250, resulting a $640 \times 128$ mel-spectrogram for 10-second audio waveform.

**Infer Time:** We benchmark the generation speed on a single A6000 GPU, and measure the average sampling time (in second) for a single 10-second sample. For all baselines and our method, we use a batch size of 1 to run the test over 15,446 test videos on VGGSound. For V2A-Mapper, since we are unable to obtain their source code for testing, we instead report the AudioLDM-L inference speed on a single sample as an approximation of infer Time of V2A-Mapper since the method is a close adaption of AudioLDM.

**CLAP score** For comparing adherence of generated audio to the text prompt, we use CLAP model to extract the audio and text embeddings, and then measure the cosine similarity between the generated audio embedding and text prompt embedding. For evaluating the video-to-audio captioning, we again apply CLAP to compute the cosine similarity between the generated audio captions and the ground truth audio for the video.

# G    Human Subjective Studies Details

For human evaluations results on audio generation shown in Table 4, we used the Amazon Mechanical Turk platform to create a survey and crowdsource responses. We evaluated 100 video samples randomly selected from VGGSound test set. We used stratified sampling such that each video comes from different audio-visual categories.

To ensure the quality of the survey, we applied constraints on accepted human raters for our survey. In particular, we selected raters that who have historical approval rate of greater than 95% as well as possess language proficiency in English. Responses that take less than 20 seconds or longer than 10 minutes are excluded from the answers. In addition, no samples could be evaluated twice by the same worker to avoid potential bias.

We set up two types of surveys: audio quality survey and audio-video relevance survey. In audio quality survey, we ask the raters to focus only on the audio quality aspect by providing the question "In Which video the overall audio quality is better?". In audio-video relevance survey, a question of "Which video whose audio is more relevant to and temporally aligned with the video." is asked. In both surveys, we ask the rater to choose their preference at a Likert scale from 1 (strong preference of baseline method) to 5 (strong preference of our method), 5 levels in total.

For each video, we request 5 responses from 5 distinct raters. To comply with the NeurIPS code of conduct and rules of the platform, we pay at a rate of 0.05 USD per each response, satisfying the lowest wage requirement in any region of the world. Upon running the evaluations on all 10 specific surveys (5 methods and each with two evaluation types), we are able to collect 500 valid responses for each pairwise comparison, from 23 distinct raters on average with $21.7 \pm 5.6$ ratings per participant.

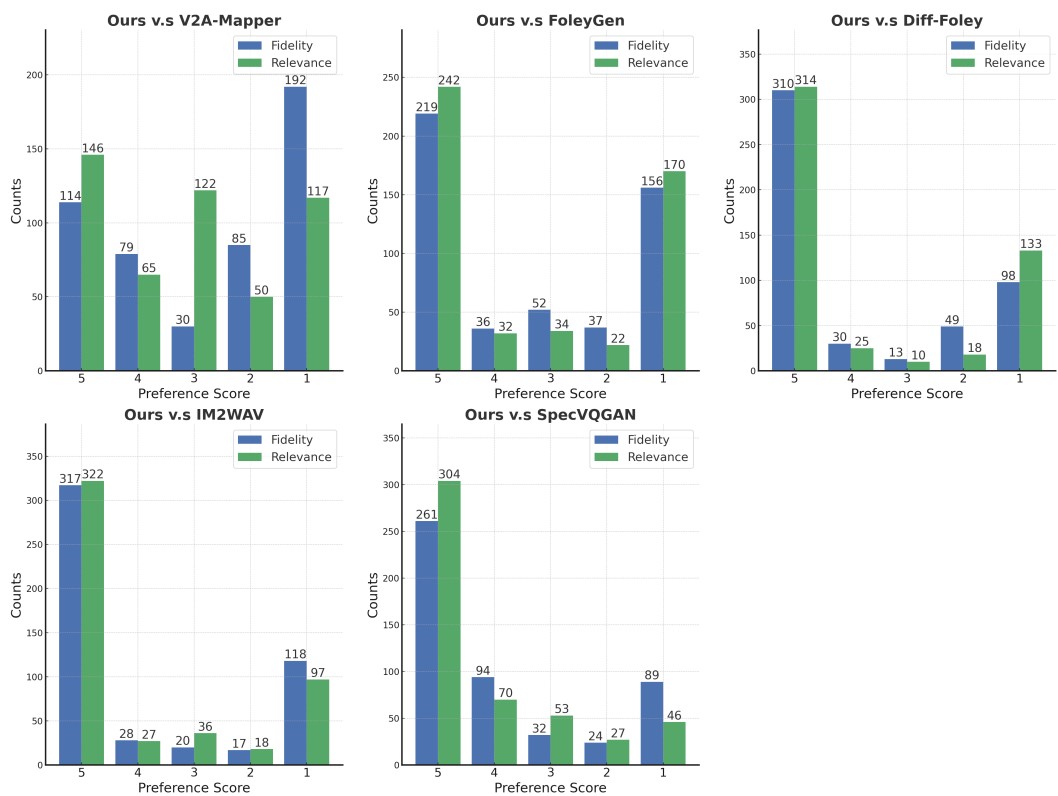

Figure 9: Details of pairwise comparison results between VATT and baseline video-to-audio methods.

We show the details of comparison results between our method and baseline methods as bar charts in Figure 9.

