# OpenReview forum: "Tell What You Hear From What You See - Video to Audio Generation Through Text"
_NeurIPS.cc/2024/Conference — NeurIPS 2024 poster_

### Official Review · Reviewer_BS9v · 2024-07-06

**Soundness:** 3
**Presentation:** 3
**Contribution:** 3
**Rating:** 6
**Confidence:** 5

**Summary:**

This paper proposes a multimodal video-to-audio generative model called **VATT** that can generate audio and audio captions given input silent videos and optional text prompts. It is capable of performing controllable text-guide video-to-audio generation and video-to-audio captioning. The framework consists of two parts:
i) **VATT converter**, an instruction fine-tuned LLM that projects video features into language space. To train such a model, the authors caption the existing audio-visual dataset by LTU-13B.
ii) **VATT audio**, a bi-directional transformer audio decoder based on **Maskgit** to generate audio tokens and text tokens with given inputs by iterative parallel decoding and Encodec as audio tokenizer.

Through the quantitative and qualitative experiments on the VGGSound dataset, they demonstrate that the proposed method achieves state-of-the-art performance in the aspect of audio quality and inference speed.

**Strengths:**

- The paper is well-written and easy to follow. The motivation is clear. All details are clearly stated in the paper and appendix.
- The paper has several technical highlights, especially on the V2A Instruction Tuning part which enables text control on the video-to-audio generation.
- The authors perform the throughout experiments and ablations to demonstrate its technical advance. The proposed method outperforms baselines on the quantitative evaluation and human study.

**Weaknesses:**

- The reviewer thinks the major contribution lies in the V2A Instruction Tuning part which enables the text control on the video-to-audio generation. While the contribution of maskgit-based audio decoder seems to be incremental.
- From the reviewer's perspective, using the VATT converter to encode visual information will lose some temporal information from the video. From the provided qualitative examples, the reviewer thinks the proposed method is not performing quite well on the generated synchronized audio but only roughly correct.

**Questions:**

The reviewer has questions about the paper:
- The reviewer wonders why Maskgit as the decoder framework. From the reviewer's side, the inference speed seems to be the main advantage.
- For the visual encoder eva-clip, it encode the feature frame-by-frame. The reviewer thinks it might lose temporal information, do authors explore this with a different encoder?
- The reported inference speed of Diff-Foley is much slower than the number reported in their paper, any clue why it is happening

**Limitations:**

Yes, the authors have adequately addressed the limitations.

---

> ### Author Rebuttal · Authors · 2024-08-06
>
> We thank the reviewer for the thoughtful review and valuable feedback. We are glad to see that the reviewer acknowledges the presentation of our work, the technical highlights and thorough experiments. We address the feedback below.
>
> **W1.** Indeed, one major contribution of our work is the V2A tuning part which allows both video-to-audio captioning as well as video-to-audio generation via text control. For the audio decoder part, our methods are inspired from previous works like MaskGIT and Soundstorm. However, we still want to emphasize our key differences from these works. MaskGIT explores only the task of class-label conditioned image generation, and it is not designed to work on cross-modal generation like video-to-audio generation. For Soundstrom, it enables text-conditioned audio generation. However, its masking designs, in particular the masking ratio distribution, are different from ours. We conducted an ablation study (see Table 5) to study how the masking distribution design affects the performance of generation quality.  The experiment could give some insights on how to design the masking ratio, which was not explored in previous works.
>
> **W2 & Q2:** We agree with the reviewer that frame-by-frame eva-CLIP features will lose fine-grained temporal details. At the current stage, we do not find any more powerful video encoder techniques to efficiently extract these fine-grained spatio-temporal features for better synchronizations. We will add this as part of the limitations in our revised manuscript, and motivate for the future research in this direction.
>
> **Q1:** We agree with the reviewer that one of the key motivations to use masking-based decoder is that it enables parallel decoding, which makes the inference speed a lot faster. The other reason we aim to use masking-based decoder is that we aim to unify different modalities using token representation, such that this kind of framework could be easily worked with multi-modal generation purposes within one model. In such cases, the choices are masking-based decoder and autoregressive decoder. For efficiency purposes again, we design our model's framework with masking-based decoder.
>
> **Q3:** For all inference speed evaluation, we perform generation with batch size = 1, while we found that Diff-Foley performs inference in a batch size of 64 and the inference time is calculated by per batch inference time divided by 64, which is the reason that their reported inference speed is significantly faster. For all other parts, we kept the same setting as theirs and used their Github repo’s notebook to perform inference. We will clarify this in our revised manuscript.
>
>
> **Flag for Ethics Review:** We obtained Institutional Review Board Approval (IRB Approval) from our institution on conducting human subject evaluations for this research project before the submission deadline. Per further request, we could supply the evidence in an anonymous way to the reviewing board committee.

---

> > ### Comment · Reviewer_BS9v · 2024-08-10
> >
> > Thanks for the authors' response. The reviewer's concerns are mostly addressed. I'd like to keep my positive score.

---

> > > ### Author Response · Authors · 2024-08-10
> > >
> > > Thank you again for your insightful comments, suggestions, and appreciation of our work! We will adjust our manuscript to reflect all the changes made in the rebuttal.

---

### Official Review · Reviewer_Gn3y · 2024-07-09

**Soundness:** 2
**Presentation:** 2
**Contribution:** 2
**Rating:** 5
**Confidence:** 4

**Summary:**

The paper introduces a two-stage model for video-to-audio generation controlled by text. In the first stage, a large language model (LLM) generates audio captions based on the video content. In the second stage, the model uses video frames and the generated audio captions to predict masked audio tokens, thereby generating the audio.

**Strengths:**

1.	An anonymous website is provided to showcase the audio generation results, clearly illustrating the effectiveness of the model.
2.	The proposed method can accomplish both video-to-audio generation and captioning tasks, and it demonstrates superior performance compared to the mentioned audio generation models.

**Weaknesses:**

1.	The content of the paper is densely packed, with many details relegated to the appendix, making it somewhat difficult to understand.
2.	There are some grammatical errors in the paper, such as in lines 133-134 where it states "showed improvement in multi-modal understanding in in tasks such as captioning and question-answering."
3.	The paper's novelty is somewhat limited. I noticed that the bidirectional-attention, Iterative Parallel Decoding, and masking design mentioned in the paper were already proposed in reference [1]. How do these differ from what is presented in this paper? If the only contribution is integrating video input into a large language model, I believe this approach alone is insufficient to support the acceptance of the paper.

[1] Zalán Borsos, Matt Sharifi, Damien Vincent, Eugene Kharitonov, Neil Zeghidour, and Marco Tagliasacchi. Soundstorm: Efficient parallel audio generation. arXiv preprint arXiv:2305.09636, 2023.

**Questions:**

1.	See the 3rd item in “Weakness”.
2.	In the quantitative evaluation of video-to-audio captioning, comparing the proposed model with LLAVA, which only accepts single image input, seems unfair. Why not compare it with large language models designed for video understanding, such as VideoChat or Video-LLaMA?

**Limitations:**

Lack of novelty, too dense content

---

> ### Author Rebuttal · Authors · 2024-08-06
>
> We thank the reviewer for the insightful feedback. We are glad that the reviewer is satisfied with the quality of provided samples and generation capability of our model, and that our model is capable of performing both video-to-audio captioning and video-to-audio generation tasks.  We address the concerns raised by the reviewer below.
>
> **W1:** We aimed to incorporate the most important experiments in the main section of our manuscript. However, we find it difficult to include all ablation studies as well as qualitative samples within the page limit due to the fact that our model works simultaneously on two tasks. We will try our best to adjust some part of the contents so that it becomes easier to read.
>
> **W2:** We will make sure to correct these grammar errors and typos in our manuscript through careful proofreading upon revision.
>
> **W3 & Q1:** In terms of technical aspect, the main differences of our proposed method from Soundstorm lie in the masking design.
>
> 1. Soundstorm only tried an arc-cosine masking ratio distribution similar to the one proposed in MaskGIT. However, we explored different masking ratio distributions during training and compared them in Table 5. The study provides critical insights regarding which masking ratio distribution would be suitable for audio generation. In general, the model performs better when the distribution has a higher mean in masking ratio. This is due to the fact that initial steps during the sampling stage are important for future decoding steps. And these initial steps are in the situations of high masking ratio. For later steps, new tokens are unmasked conditioned on more clues as the masking ratio goes lower, the generation becomes less challenging, thus making the learning at lower masking ratio easier during the training.
>
>
> 2. Soundstorm designed a complicated masking pattern, where any selected token will be masked together with tokens at higher levels than it. While our approach does not require this pattern dependencies and still performs well.
>
> We will highlight these technical similarities and differences in our revised manuscript.
>
> Aside from directly comparing masking novelty of VATT, we would like to reiterate that additional key novelty in our work is integrating LLM into the audio generation pipeline which enables the capability of suggesting an audio for the video through captioning, as well as generation of audio from video with text feedback. This cannot be achieved by mere combination of existing approaches since the ability to produce text as well as encode text as a condition for the output of another modality like audio is not achievable using these commonly used off-the-shelf text encoders like BERT or T5. Our work is the first to leverage the capability of LLM to achieve video-to-audio generation through text. Such a unique design boosts interactivity via text control as well as provides interpretability of the model by providing captions, which we believe is an important novelty of our method. As mentioned by Reviewer BS9v, “The paper has several technical highlights, especially on the V2A Instruction Tuning part which enables text control on the video-to-audio generation.”
>
>
> **Q2:**  We further compare our method against existing video LLM models. We experiment with Video-LLAMA as suggested by the reviewer in two ways:
>
> 1. We conduct zero-shot captioning using Video-LLAMA to generate audio captions for the VGGSound test dataset. We tried different prompts and ended up using “User/ What sounds could match the video?” as it allows the model to follow the instruction closely.
>
> 2. Because Video-LLAMA has not been trained on VGGSound dataset and LTU generated captions, we implement a similar structure of Video-LLAMA and then train it on our LTU-generated captioning data. Specifically, we replaced the BLIP-2 visual features with our eva02-CLIP-L visual features due to the expensive pre-processing time for all videos in VGGSound and AudioSet data. For the Video-QFormer component of Video-LLAMA, we keep it the same as Video-LLAMA, and we name this model as VATT-Qformer LLama.
>
> As shown in Table 1R3, Video-LLAMA’s zero-shot video-to-audio captioning performance stays close to LLAVA albeit utilizing all of the video frames. When we train a similar structure to Video-LLAMA from scratch on LTU captions, we find that its performance is close to VATT-Converter LLama. The linear projection adopted by us and the Qformer projection methods don’t make much difference in our video-to-audio captioning task, and we choose a simpler way of projection.
>
> Table 1R3: Comparison of video-to-audio captions on NLG evaluation metrics and text-audio relevance (CLAP Score). Bold denotes the new results.
> | Methods                    | BertScore (F1) ↑ | BLEU-4 ↑ | ROUGE-L ↑ | CIDEr ↑ | CLAP Score ↑ |
> |----------------------------|------------------|----------|-----------|--------|--------------|
> | LLAVA w/ Visual Prompt     | 0.855            | 0.089    | 0.137     | 0.026  | 0.213        |
> | LLAVA w/ Audio Prompt      | 0.870            | 0.123    | 0.155     | 0.095  | 0.182        |
> | **Video-LLAMA w/ Audio Prompt** | **0.861**    | **0.091** | **0.117**| **0.021** | **0.204** |
> | VATT Converter - Gemma     | 0.900            | 0.345    | 0.337     | 0.926  | 0.229        |
> | **VATT-Qformer - LLama**   | **0.907**        | **0.419** | **0.375** | **1.264** | **0.245** |
> | VATT Converter - LLama     | 0.909            | 0.424    | 0.384     | 1.354  | 0.263        |

---

> > ### Comment · Reviewer_Gn3y · 2024-08-10
> >
> > The author's response basically solved my confusion, so I chose to boost my score to borderline accept.

---

> > > ### Author Response · Authors · 2024-08-10
> > >
> > > Thank you so much for your valuable feedback and time devotion for reviewing the work along with the rebuttal! We will incorporate these modifications in our revised manuscript.

---

### Official Review · Reviewer_rzvn · 2024-07-12

**Soundness:** 2
**Presentation:** 3
**Contribution:** 2
**Rating:** 5
**Confidence:** 5

**Summary:**

In this paper, the authors propose VATT, a multi-modal generative framework for text-guided video-to-audio generation. VATT consists of two key modules VATT Converter and VATT Audio. The former component maps video features to LLM vector space with a projection layer. The latter one generates audio tokens with a bi-directional transformer. Experiments on datasets VGGSound and AudioSet-2M demonstrate that the proposed VATT framework surpasses existing SOTA in terms of KLD score.

**Strengths:**

1. The authors identified limitations within the existing video-to-audio generative modeling techniques, indicating a clear understanding of the challenges in the field.
2. The publication of the generated samples is also commendable.

**Weaknesses:**

1. As stated by the authors, the diversity of text generated by LLM could influence the generated audio. However, it is unclear how the quality of the generated caption affects the results. It is suggested that the authors include an ablation study comparing the results of different captions quality and choose some examples to analyze the reasons.
2. In the section on quantitative results, the proposed method is only evaluated on VGGSound, which is not enough to prove the generalization and robustness of the proposed method. It would be more convincing to conduct experiments and ablation studies on more datasets.
3. There seems to be a writing error in part D of Appendix, “we use existing an existing audio large language model…”

**Questions:**

See weaknesses

**Limitations:**

yes

---

> ### Author Rebuttal · Authors · 2024-08-06
>
> We thank the reviewer for thoughtful feedback and acknowledging the quality of generated samples from VATT. We address the feedback and concerns below.
>
> **W1:** The quality of generated captions do affect the generation quality of the model in the V+T -> A stage. We conduct an additional ablation study by providing captions generated by different models as inputs to the same VATT model, i.e., VATT-LLama-T or VATT-Gemma-T to evaluate the impact of models’ captioning ability on the audio generation ability. We compare two generated captions from VATT-Converter - Gemma and VATT Converter - LLama. For reference, we also report the result of ground truth audio caption as inputs to the model. As shown in the Table 1R2, when the same model is fed with generated captions with different quality, the model’s audio generation performance are different. This effect is already expected from “Table 6: Ablation on self-prompting text-guided generation” in our paper. Combined with results in Table 3 in our paper, we can see that as the caption quality improves, the text-conditioned video-to-audio generation performance also improves, in particular from the FAD and KLD scores. Notably, the ground-truth audio captions generated by LTU has the highest CLAP score (measured with respect to the ground truth audio in original video) of 0.379 as stated in the paper, reflecting the best caption quality. Feeding such ground truth captions as input to the model also leads to the best audio generation results.
>
> As for the qualitative analysis on how the captions affect the audio generation quality, we select an example video with YouTube ID “j1kM-hC44Ok_000002.mp4” from VGGSound test set to analyze. The caption generated by VATT-LLama Converter is “A car is driving and crushing.”, while the caption generated by VATT-Gemma is “A car is driving and a man speaks with wind noise in the background.” As can be seen from the video, the car is crushing over some objects and making cracking sounds. The former caption is better aligned with the visual scene, and it generates the sound that matches closely to what is intended. While the latter caption is unrelated to the key event happening in the video, leading to poorer results. We will incorporate qualitative examples as well as demos on our website, and make more detailed analysis in our revised manuscript to help understand the correspondence between the caption quality and the generated audio.
>
> From another perspective, our design by enabling the model to take an extra caption as input is a useful feature since the user can provide reasonable text input to steer the generated sound with variations to some extent. In this aspect, our qualitative samples from the provided link in the paper as well as Figure 5 in the Appendix C demonstrate some interesting scenarios where a user provides different plausible captions for the model to generate.
>
> Table 1R2: The impact of caption quality on V + T -> A performance.
>
> | Methods        | Text Prompt             | KLD ↓ | FAD ↓ | Align Acc ↑ |
> |----------------|-------------------------|-------|-------|-------------|
> | VATT-Gemma-T   | VATT-Converter Gemma    | 2.40  | 3.67  | 80.07       |
> | VATT-Gemma-T   | VATT-Converter LLama    | 2.26  | 3.20  | 80.42       |
> | VATT-Gemma-T   | Ground Truth            | 1.66  | 2.98  | 81.48       |
> | VATT-LLama-T   | VATT-Converter Gemma    | 2.57  | 2.67  | 79.20       |
> | VATT-LLama-T   | VATT-Converter LLama    | 2.38  | 2.58  | 80.41       |
> | VATT-LLama-T   | Ground Truth            | 1.41  | 2.54  | 80.16       |
>
>
> **W2:** We use VGGSound as our main dataset and benchmark to evaluate since it is not only a large-scale audio-visual dataset with around 200K videos across many categories, but also the quality of audio-visual alignment is quite good. To further show the generalization capability of VATT, we experiment with AudioCaps dataset. Due to limited video samples in AudioCaps, we finetuned our VGGSound pretrained VATT model on AudioCaps dataset in two settings, with and without text prompts. To keep the comparison fair, we use the ground truth audio captions from AudioCaps as the text prompts. We use VATT-LLama and VATT-LLama-T to compare against AudioGen and AudioLDM-2. As shown in Table 2R2, VATT-LLama-T performs on a similar level to AudioGen in terms of FAD and KLD score, while falling behind AudioLDM-2. It is noteworthy that the audio decoder of both AudioGen and AudioLDM-2 are pretrained on much larger data scale (7000 hrs and 30000 hrs audio respectively) than ours (700 hrs audio). Despite this, our method still performs reasonably well on this dataset. We plan to conduct further  ablation studies by scaling up the training by incorporating more data to train VATT on the video-to-audio generation task.
>
> Table 2R2: Quantitative results against text-to-audio generation methods on AudioCaps test set.
>
> | Methods       |  KLD ↓ |  FAD ↓ | Align Acc ↑ | CLAP Score ↑ |
> |---------------|------|------|-----------|------------|
> | AudioGen      | 2.09 | 3.13 | 58.26     | 0.447      |
> | AudioLDM-2    | 1.64 | 1.86 | 60.32     | 0.432      |
> | VATT-LLama    | 2.53 | 3.42 | 75.76     | -          |
> | VATT-LLama-T  | 2.07 | 3.25 | 74.89     | 0.376      |
>
> **W3:** We will correct the grammar errors and typos like this in our revised manuscript through careful proofreading.

---

> ### Author Response · Authors · 2024-08-12
>
> Dear Reviewer,
>
> We are very thankful for your valuable feedbacks and comments! Let me know if you have further questions regarding the rebuttal response or any other questions related to the paper. We look forward to your further feedbacks!
>
> Thanks,
>
> Authors of Paper Submission 1915

---

> > ### Author Response · Authors · 2024-08-13
> >
> > Since the discussion period ends soon, we want to reach out to see if you have any further advice or feedback and whether there is more information we could provide for reconsideration of your score? Please let us know and we will be happy to engage in further discussions. Thanks!

---

### Official Review · Reviewer_5yrj · 2024-07-13

**Soundness:** 3
**Presentation:** 3
**Contribution:** 3
**Rating:** 6
**Confidence:** 4

**Summary:**

- The paper proposes a model for video-to-audio generation (main task) and video-to-audio captioning (auxilliary task).
- The text is optionally used to control audio generation for ambiguous video cases.
- They use two step approach i.e. video-to-caption stage and video+text-to-audio stage
- Stage 1: video-to-caption
  - This stage basically converts videos and text to audio relevant captions
  - They train a VATT converter that uses LoRA finetuning to get audio captions.
  - This stage uses AudioSet and VGGSound and captions generated using LTU model.

- Stage 2: video+text-to-audio
  - This stage generates audio tokens, given video+text as input. The video+text is converted to audio relevant captions using stage-1
  - They use LM based audio token generation. The VATT Audio decoder learns to predict the masked audio tokens. During inference, iterating parallel decoding is used.
  - VGGSound is used to train this stage.

- There model shows comparative and better performance on video-to-audio(V2A) generation task and text to audio generation task. Add text to V2A task further improves KLD score.
- Stage 1 when compared to baselines can perform better on video-to-audio captioning.
- Qualitative and human evaluation shows preference towards their work.

**Strengths:**

- The paper proposes a new problem of utilizing text for video-to-audio generation.
- Several novel and interesting technical contributions. i.e.
 - Video+text conditioning to generate audio by converting this condition to joint audio-caption.
 - Using synthetic captions for training and LoRA for MLLM finetuning.
 - Using iterative parallel decoding and bi-directional self-attention.
- Their approach improves on video-to-audio generation, and text-to-audio generation both quantitatively and qualitatively.
- The results also show improvement in video-to-caption generation.

**Weaknesses:**

- Text captions generated by LTU are used as ground truth for video-to-audio generation, text-to-audio generation, and video-to-audio captioning. The authors should verify the correctness of the captions.

- Table 2 results for Text-to-audio generation seem slightly unfair for the baselines.
      - The LTU-generated caption may have some biases or patterns of caption generation that VATT has seen during training while the
         baselines have not.
      - Though AudioGen has seen VGGSound tags, it has not seen the captions generated by LTU (which VATT has).
      - AudioLDM2 has not even seen VGGSound data for training.
      - A fair comparison would be finetuning/training from scratch Text-to-audio models on VGGSound with LTU-generated captions. Or VATT should be compared on AudioCaps (with videos downloaded from YouTube).

- Table 3 results are also a little unfair:
  - Similar to the argument above LLAVA has not seen LTU-generated captions for VGGSOUND.
  - Also LLAVA takes only 1 frame while VATT takes more frames.
  - Maybe a fair comparison would be to train some captioning models on LTU-generated data.

Minor comments:
- The captions for the tables should come above the table. For eg. Table 1,2,3
- Missing reference: A previous work by Mo. et al [1] that utilizes text+video-to-audio generation. This should be mentioned
- Align-acc for diff-foley is 82.47 in the paper vs 94.05 in the original paper

[1] DiffAVA: Personalized text-to-audio generation with visual alignment

**Questions:**

My questions are based on the arguments mentioned in the weakness section:

- Can the correctness of the LTU-generated captions be verified? Since they are used as the ground-truth for several tasks?

- Can the baseline text-to-audio(TTA) generation models be trained on VGGSound+LTU captions? Can VATT be compared on AudioCaps?

- Can VATT be compared on other video captioning datasets? Or the existing video-captioning models be finetuned/trained from scratch on this LTU data?

- Why are some metrics different from the numbers quoted in the original paper? For eg. Diff-foley alignment score.

**Limitations:**

Yes the authors adequately addressed the limitations.

---

> ### Author Rebuttal · Authors · 2024-08-06
>
> We thank the reviewer for their insightful review and valuable feedback. We appreciate the reviewer feedback that our proposed methods include novelties and technical contributions. We address the raised concerns below.
>
> **W1 & Q1:** We manually verified the validity of LTU-generated captions prior to using them as synthetic GT. As a result of reviewer feedback, we performed an experiment to further evaluate its correctness. We randomly selected 100 videos from VGGSound test set with stratified sampling according to video categories to conduct a human study. We use the 1-5 point MOS (Mean-Opinion-Score) scale (the higher the better) to measure correctness of the captions. We provide pairs of videos and the corresponding captions to the raters, asking “How accurately the provided caption reflects the sound events happening in the video? **1. Inaccurate and irrelevant.** 2. Relevant but inaccurate with many mistakes. 3. Partially accurate but missing details and with mistakes. 4. Mostly accurate with some minor mistakes. **5. Accurate and complete.**” We used the MTurk platform to perform the evaluation and collected a total of 300 responses. The generated captions have a high MOS of mean 4.72 and std 0.37, providing another indication for the validity of the synthetic ground truth.
>
>
> **W2 & Q2:** We thank the reviewer for asking regarding further comparison against existing text-to-audio methods. Due to the time constraints of the rebuttal, we chose to finetune our VATT model on AudioCaps benchmark instead of training AudioGen and AudioLDM-2 using LTU generated captions from scratch. Specifically, due to the limited data size of AudioCaps, we use our VGGSound pretrained checkpoint to finetune instead of training on AudioCaps from scratch. As shown in the Table 1R1, VATT achieves similar performance against AudioGen on FAD and KLD metrics, while still showing some gaps from AudioLDM-2. One clear reason is that AudioGen and AudioLDM-2 are pretrained on much larger audio-text data (7000 hrs and 30000 hrs audio respectively) than ours (700 hrs audio). In addition, we also find that the audio-visual alignment from AudioCaps dataset (videos from AudioSet) is not as good as VGGSound, often suffering from static frames and weak correspondence of audio and visual. Therefore, the visual condition is not as effective as the VGGSound videos.
>
>
> Table 1R1: Quantitative results against text-to-audio generation methods on AudioCaps test set.
> | Methods       |  KLD ↓ |  FAD ↓ | Align Acc ↑ | CLAP Score ↑ |
> |---------------|------|------|-----------|------------|
> | AudioGen      | 2.09 | 3.13 | 58.26     | 0.447      |
> | AudioLDM-2    | 1.64 | 1.86 | 60.32     | 0.432      |
> | VATT-LLama    | 2.53 | 3.42 | 75.76     | -          |
> | VATT-LLama-T  | 2.07 | 3.25 | 74.89     | 0.376      |
>
>
> **W3 & Q3:** We appreciate the reviewer pointing to the need for further comparison on video-to-audio caption tasks. We have added two additional models/setups in the comparison.
>
>
> To address the concerns that LLAVA is only trained on single frames rather than the whole video, we instead use an existing video LLM , Video-LLAMA-7B, to perform zero-shot video-to-audio captioning. Specifically, following the Video-LLAMA’s Github repository instructions, we directly input the VGGSound videos into the VL branch of the model, and prompt it to generate audio captions using the instruction “User/ What sounds could match the video?”
>
> Since Video-LLAMA is still not pretrained on VGGSound dataset and LTU generated captions, we implement a similar structure of Video-LLAMA and train on our LTU-generated captioning data. We replaced the original BLIP-2 visual features used by Video-LLAMA with our eva02-CLIP-L visual features due to the expensive pre-processing time for all BLIP-2 features from videos in VGGSound and AudioSet. For the Video-QFormer component of Video-LLAMA, we keep it the same as Video-LLAMA, and we name this model as VATT-Qformer - LLama.
>
> As shown in Table 2R1, zero-shot performance of Video-LLAMA is similar to LLAVA, while VATT-Qformer - LLama trained from scratch performs very close to VATT Converter - LLama. It is note-worthy that the only difference between VATT Converter - LLama and VATT-Qformer - LLama is the projection method, and we find that the linear projection adopted by VATT Converter  is enough for the task to perform well.
>
> Table 2R1: Comparison of video-to-audio captions on NLG evaluation metrics and text-audio relevance (CLAP Score). Bold denotes the new results.
>
> | Methods                    | BertScore (F1) ↑ | BLEU-4 ↑ | ROUGE-L ↑ | CIDEr ↑ | CLAP Score ↑ |
> |----------------------------|------------------|----------|-----------|--------|--------------|
> | LLAVA w/ Visual Prompt     | 0.855            | 0.089    | 0.137     | 0.026  | 0.213        |
> | LLAVA w/ Audio Prompt      | 0.870            | 0.123    | 0.155     | 0.095  | 0.182        |
> | **Video-LLAMA w/ Audio Prompt** | **0.861**    | **0.091** | **0.117**| **0.021** | **0.204** |
> | VATT Converter - Gemma     | 0.900            | 0.345    | 0.337     | 0.926  | 0.229        |
> | **VATT-Qformer - LLama**   | **0.907**        | **0.419** | **0.375** | **1.264** | **0.245** |
> | VATT Converter - LLama     | 0.909            | 0.424    | 0.384     | 1.354  | 0.263        |
>
>
> **Minor Comments 1:** We will move the table titles and captions above the tables to follow the NeurIPS format.
>
> **Minor Comments 2:** We will make sure to cite the reference mentioned by the reviewer as it is closely related to our work.
>
> **Minor Comments 3 & Q4:** We did find differences between our experiment results and the ones reported in Diff-Foley’s results. However, we strictly follow the guidelines and instructions from their official repository to perform audio generation on the VGGSound test set, and we could not achieve their claimed performance. We provided all the necessary implementation details for the baseline comparisons in Appendix F for reference.

---

> > ### Comment · Reviewer_5yrj · 2024-08-10
> >
> > Thanks for the comprehensive explanation of my questions. Hence, I would keep my positive score.
> > Please include the additional experiments and clarifications in the main text or appendix of the paper.

---

> > > ### Author Response · Authors · 2024-08-10
> > >
> > > Thank you for your detailed and thoughtful reviews, and appreciation of our work! We will include all the mentioned items in the rebuttal into our revised manuscript properly.

---

### Decision · Program_Chairs · 2024-09-25

**Decision:**

Accept (poster)

**Comment:**

This paper studies the task of video-to-audio generation task. The authors propose to additionally include text conditioning to improve and stabilize the generation process. The authors achieve this joint audio generation and captioning framework.
The authors addressed the reviewers comments during the rebuttal phase, while providing additional results and clarifications. I highly encourage the authors to include these in the final manuscript.
There are several missing references such as:
[1] Mo, Shentong, Jing Shi, and Yapeng Tian. "DiffAVA: Personalized text-to-audio generation with visual alignment." arXiv preprint arXiv:2305.12903 (2023).
[2] Ziv, Alon, et al. "Masked audio generation using a single non-autoregressive transformer." arXiv preprint arXiv:2401.04577 (2024).